# A robust gene-stacking method utilizing yeast assembly for plant synthetic biology

Patrick M. Shih[1,2], Khanh Vuu[1,2], Nasim Mansoori[1,2], Leïla Ayad[1,2], Katherine B. Louie[3,4], Benjamin P. Bowen[3,4], Trent R. Northen[1,3,4] & Dominique Loqué[1,2,5]

The advent and growth of synthetic biology has demonstrated its potential as a promising avenue of research to address many societal needs. However, plant synthetic biology efforts have been hampered by a dearth of DNA part libraries, versatile transformation vectors and efficient assembly strategies. Here, we describe a versatile system (named jStack) utilizing yeast homologous recombination to efficiently assemble DNA into plant transformation vectors. We demonstrate how this method can facilitate pathway engineering of molecules of pharmaceutical interest, production of potential biofuels and shuffling of disease-resistance traits between crop species. Our approach provides a powerful alternative to conventional strategies for stacking genes and traits to address many impending environmental and agricultural challenges.

[1] Joint BioEnergy Institute, Emery Station East, 5885 Hollis St, 4th Floor, Emeryville, California 94608, USA. [2] Biological Systems and Engineering Division, Lawrence Berkeley National Laboratory, 1 Cyclotron Road, Berkeley, Berkeley, California 94720, USA. [3] Environmental Genomics and Systems Biology Division, Lawrence Berkeley National Laboratory, 1 Cyclotron Road, Berkeley, Berkeley, California 94720, USA. [4] Joint Genome Institute, 2800 Mitchell Drive, Walnut Creek, California 94598, USA. [5] Université Claude Bernard Lyon 1, INSA de Lyon, CNRS, UMR5240, Microbiologie, Adaptation et Pathogénie, 10 rue Raphaël Dubois, F-69622 Villeurbanne, France. Correspondence and requests for materials should be addressed to D.L. (email: dloque@lbl.gov).

I n the same manner that circuits and electronics are rationally assembled with parts and devices, synthetic biology aims to introduce engineering principles into molecular biology by assembling DNA parts together to reprogram and repurpose organisms. Already, humans have bred and refashioned plants for our own purposes through thousands of years of domestication. Introducing concepts of synthetic biology into plants will dramatically advance our ability to more rapidly and precisely modify crops beyond traditional methods.

Although synthetic biology may provide the solution to many agricultural challenges, the development of synthetic biology for plants is still in its infancy in contrast to that in the microbial field. Microbes, such as yeast and *Escherichia coli* (*E. coli*), have received much of the attention in developing synthetic biology tools due to their fast generation time and the ease of working with these organisms in laboratories, whereas little research has been pursued to advance synthetic biology in plants. A shortage of characterized DNA parts, along with the difficulty of efficiently assembling multiple and large fragments of DNA into plant transformation vectors, has limited progress in studying and engineering plants to the same degree as their microbial counterparts. Consequently, the majority of plant biotechnology efforts have predominantly targeted single to very few genes—severely limiting the capacity to modify efficiently desired traits. The development of basic tools, resources and methods will provide the backbone for plant synthetic biology. In an effort to expedite plant research and crop engineering, we have developed an efficient and affordable method for DNA assembly into a suite of plant transformation vectors leveraging *in vivo* yeast homologous recombination, along with a library of over a 100 publicly available DNA parts.

## Results

**Developing plant DNA-stacking strategies with yeast assembly.** One hurdle to the progress of plant synthetic biology is the capacity for rapid, flexible and larger-scale DNA assembly into plant expression vectors. One technique that has proven to be robust is the use of *in vivo* yeast homologous recombination to efficiently organize and stitch together large fragments of linearized DNA, ranging from a few kilobases (kb) to whole bacterial genomes[1,2]. To utilize this technique, we have modified various plant binary vectors to be compatible with yeast replication and selection systems, generating a suite of yeast-compatible binary (pYB) vectors (Supplementary Fig. 1). DNA fragments are assembled into pYB vectors through overlapping homologous sequences, which recombine through designated homology arms on the pYB vector backbone. Because multiple DNA fragments can easily be assembled simultaneously, pYB vectors provide a novel method for rapid assembly of multiple DNA fragments and genes.

The term 'gene stacking' may refer to a number of strategies to assemble combinations of genes/alleles together, such as (1) crossing/breeding two traits from separate parent strains into one line, (2) successively transforming single genes into one host to introduce multiple genes in a piecemeal approach and (3) transiently expressing multiple genes using an *Agrobacterium*-mediated transformation method where multiple *Agrobacterium* strains are used and each strain contains plasmids to express a single gene. In this study, we refer to gene stacking as assembling multiple gene cassettes into the transfer DNA region (T-DNA) of a binary plasmid to simultaneously deliver multiple genes into a single locus in the host plant genome in one transformation event, enabling cleaner transgenic events where all transgenes are physically linked together. Given the intent of this DNA assembly platform to stack DNA and genes and the institution in

which it was developed (Joint BioEnergy Institute) we have correspondingly named our jStack.

Although the versatility of jStack permits scientists to freely choose from an array of molecular biology techniques (for example, PCR amplicons, Gibson assembly, DNA synthesis, endonuclease-based cloning) to generate the fragments that will ultimately be recombined into pYB vectors, the importance of having standardized and universal components cannot be understated. In engineering disciplines, the standardization of parts has enabled collaboration through common components and innovation using existing devices. Building on efforts to establish a standard syntax in plant synthetic biology[3], we have developed a hierarchical scheme for the assembly of genes and DNA parts into pYB vectors. Importantly, our technique is compatible with other existing methods that use Golden Gate cloning[4,5], thus maintaining the ability to collaborate and share characterized DNA parts. Importantly, yeast homologous recombination-based DNA assembly has proven to be a robust method for large-scale assembly and compatible with other methods[1,6,7]. Other DNA assembly methods may face challenges when scaling to larger DNA assemblies, as best demonstrated in whole genome synthesis efforts. Moreover, a combination of several methods (for example, Gibson assembly or endonuclease-based cloning) with yeast homologous recombination-based DNA assembly can enable larger assemblies, as Gibson and colleagues leveraged yeast assembly for the final stitching together of an entire bacterial genome, while relying on restriction endonucleases for the construction of smaller DNA fragments[1].

Cloning can be a mundane and cumbersome aspect of plant molecular biology; to expedite the process, we have built a two-tier system to streamline DNA assembly. First, Level 1 functional gene cassettes (Fig. 1a), composed of a linker, promoter, coding sequence and terminator, are assembled via Type IIS restriction enzymes. Next, in Level 2, various functional gene cassettes are assembled together in order by their linker sequences via yeast *in vivo* homologous recombination (Fig. 1b). To facilitate Level 1 assembly, a green fluorescent protein (GFP) dropout-cassette is lost with the successful assembly of a complete gene cassette into the backbone, enabling green/white selection when colonies are exposed to ultraviolet light. Furthermore, because cloning difficulties can arise owing to *E. coli* toxicity issues, Level 1 assembly can be cloned using one of two intermediate vectors which differ in *E. coli* plasmid copy number (Supplementary Table 1). Level 1 gene cassettes are designed to contain homologous regions between terminator and linker regions to enable *in vivo* yeast homologous recombination and ordered assemblies. Next, in Level 2, all pYB vectors are linearized to release a URA3 dropout-cassette, which will be replaced with the recombined cassettes of interest. Transformed yeast are plated onto plates containing 5-Fluoroorotic acid which kills yeast that still have retained the URA3 cassette, thus selecting for assembled constructs. The use of these selection schemes enables robust stacking of multiple cassettes simultaneously.

**Building an open-source library of plant-specific DNA parts.** Synthetic biology is based on the design, manipulation and assembly of DNA parts; fittingly, the characterization and standardization of these parts is a prerequisite for nearly all synthetic biology endeavors. As a result, the greater synthetic biology community has made efforts to standardize and make DNA parts available to the community through catalogues and part registries; however, the vast majority of these parts are for microbial hosts. The BioBrick standardization of DNA assembly is by far the most widespread, and of the 35,000 BioBrick biological components in the AddGene repository, less than 0.7%

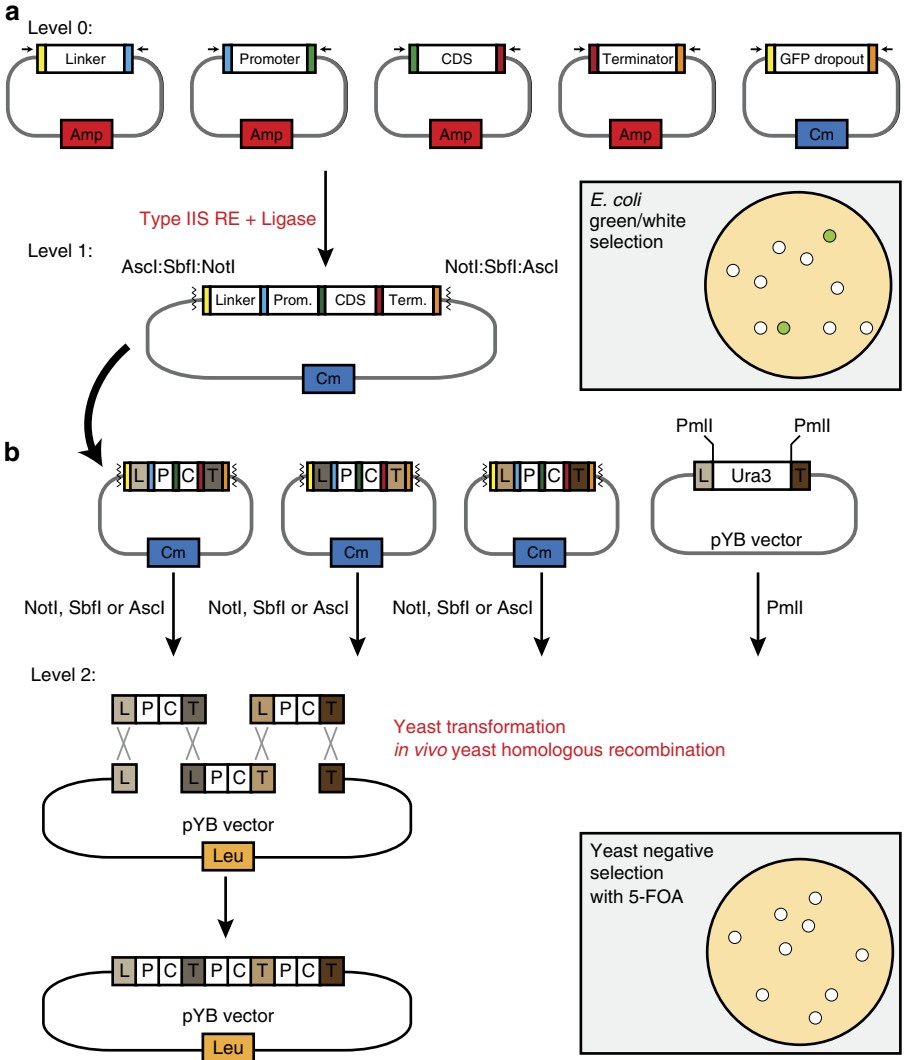

**Figure 1 | jStack hierarchical assembly.** (**a**) A library of Level 0 parts composed of linkers (linker, L), promoters (Prom., P), coding sequences (CDS; C) and terminators (Term., T) are available. Level 1 assembly consists of digesting desired Level 0 parts along with their intermediate vector backbone (containing a complementary antibiotic selection; chloramphenicol, Cm) with a Type IIS restriction enzyme followed by ligation facilitated by compatible sticky ends. Type IIS enzymes generating compatible sticky ends are denoted by the tiny black arrows above Level 0 plasmids. A GFP dropout-cassette is cut out from the intermediate vector backbone allowing for green/white selection, where all the white colonies have correctly assembled Level 1 constructs (illustrated in top right grey box). Compatibility of liberated ends by the Type IIS restriction enzymes (illustrated with coloured box at the end of each DNA part) allows assembling of all DNA parts in the correct order. (**b**) Various Level 1 constructs with compatible linker and terminator sequences that have homology to one another are digested with one of three flanking rare Type II restriction enzyme cutsites (NotI, SbfI or AscI). These are all transformed into yeast with a PmlI linearized pYB vector, which loses the Ura3 dropout-cassette. Homology between linker and terminator sequences allows DNA assembly via *in vivo* yeast recombination in the correct order, circularizing the pYB vector. The loss of the Ura3 cassette and the presence of a Leu2 marker on the backbone confer the resistance to 5-FOA and autotrophy for leucine, respectively; thus, any colonies that grow on minimal SD-Leu supplemented with 5-FOA, have recombined the various DNA fragments together into the pYB vector.

are plant-specific[8]. In comparison with workhorse microbes such as *E. coli* and yeast, the higher level of complexity displayed in plants intrinsically necessitates a larger and more concerted effort in characterizing plant-specific DNA parts. For example, DNA parts for plant promoters may need to be specific to a given cell type or in response to a certain environmental stimulus. Although these challenges are not relevant or easier to address in unicellular microbes, the majority of DNA parts have been characterized and deposited for bacteria. Hence, there is a need for a concerted effort in standardizing and depositing DNA parts for plant synthetic biology.

In an effort to address the paucity of DNA parts for plant synthetic biology, we have generated a library of 115 plant-specific DNA parts (Level 0) including commonly used promoters, genes and terminator parts, which are publicly available through the ICE repository (Supplementary Table 2)[9]. To enable consistency with other existing Golden Gate cloning-based methods, we have designed compatible Type IIS restriction enzymes overhang into our DNA parts. One notable limitation of existing methods utilizing hierarchical Golden Gate cloning is the need to utilize 'domesticated' parts[4,5]; where all Type IIS restriction enzyme cutsites are abolished to facilitate assembly, but may potentially affect functionality of DNA parts (for example, promoters). To address this issue, our method is capable of using any Type IIS restriction enzymes, thus avoiding the rigidity of these previously described methods. This is made

possible by the fact that our method necessitates only one assembly level, which uses Type IIS restriction enzyme cloning. As a demonstration of the full pipeline from library parts to stable expression of a jStack assembly in crops, we generated transgenic soybean roots expressing three stacked reporter genes (Fig. 2a, Supplementary Table 3).

**Utilizing yeast assembly for plant metabolic engineering.** One clear application of gene stacking is in advancing plant metabolic engineering efforts. The introduction and optimization of complex metabolic pathways into plants opens new

opportunities, from introducing agriculturally relevant traits to utilizing plant-based platforms for the production of molecules of interest. One biotechnologically relevant metabolism is terpenoid biosynthesis, owing to its potential in providing advanced biofuels[10], pest control[11] and pharmaceuticals[12]. As a proof of concept, we set out to produce the molecule bisabolene, a precursor to bisabolane and an alternative to D2 diesel fuel[10]. The heterologous production of bisabolene in plants requires only one enzyme, bisabolene synthase, using farnesyl diphosphate (FPP) as substrate. However, with an increased capacity for DNA assembly, it becomes trivial to rapidly test the additive effects of other stacked genes (Supplementary Table 3). First, we

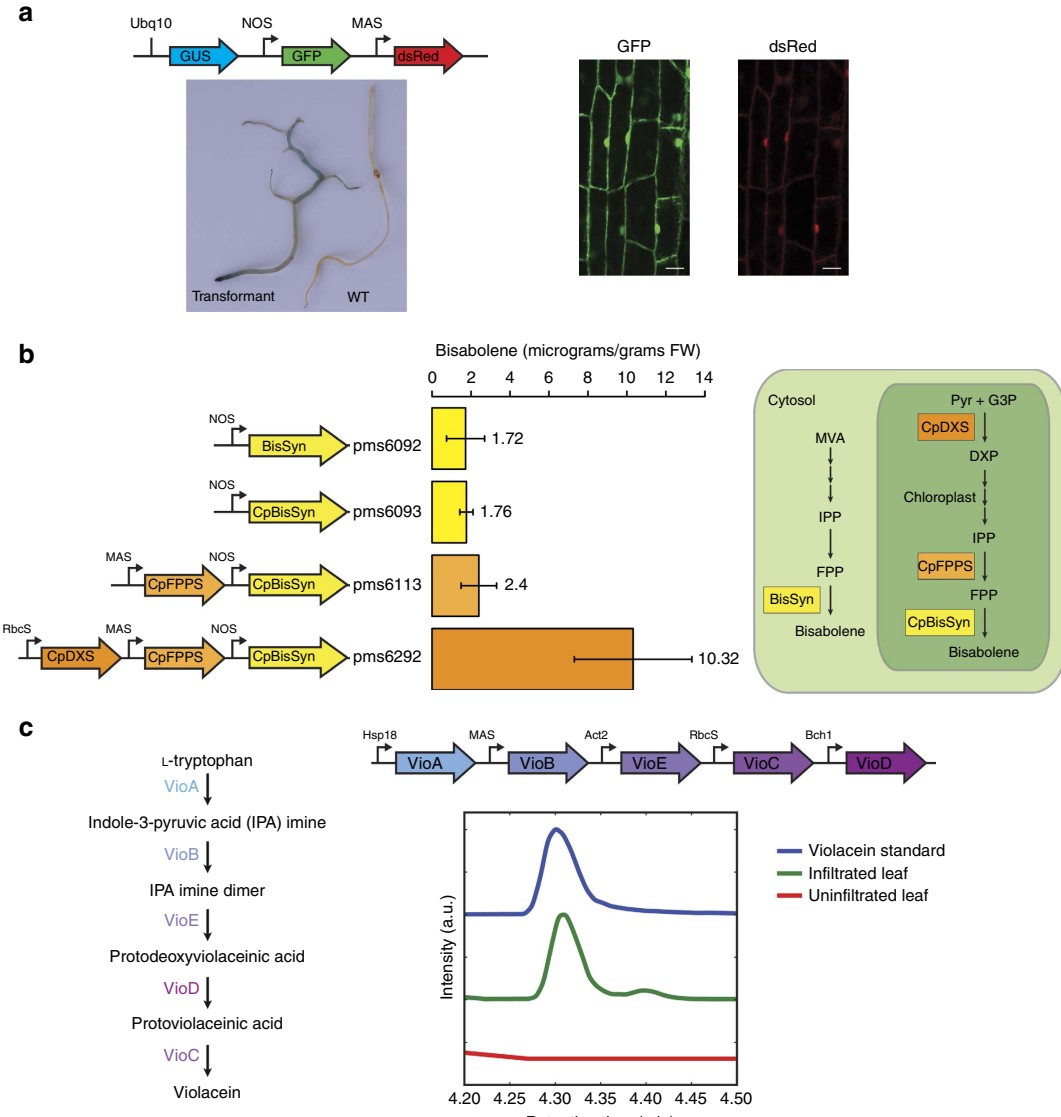

**Figure 2 | Assembling multi-gene pathways into pYB vectors with jStack.** (**a**) Stacking various reporter genes together into one transfer DNA: β-glucuronidase (GUS), GFP, DsRed. Promoters driving each gene are labelled and denoted by black arrows in front of coding sequences. Stable soybean root transformant lines were generated via *Agrobacterium rhizogenes*. Transformed roots exhibited GUS activity and GFP and DsRed fluorescence. Scale bar represents 20 μm. (**b**) Optimizing engineered metabolic pathways of a potential biofuel with gene stacking. Stacking various upstream enzymes in conjunction with new organellar targeting increased Bisabolene yields *in planta*—values are labelled in the graph. By stacking upstream enzymes such as chloroplast-localized farnesyl pyrophosphate synthase (CpFPPS) and chloroplast-localized 1-Deoxy-D-xylulose 5-phosphate synthase (CpDXS) with either cytosolic or plastid-targeted bisabolene synthase (BisSyn and CpBisSyn, respectively), increased yields of bisabolene were observed. Error bars indicate standard deviation, $n = 3$. Mevalonate (MVA), isopentenyl pyrophosphate (IPP), farnesyl pyrophosphate (FPP), pyruvate (Pyr), glyceraldehyde 3-phosphate (G3P). (**c**) Engineering heterologous bacterial metabolites into plants. The cytotoxic molecule violacein is found in various soil bacteria and requires five enzymes (VioA, VioB, VioC, VioD and VioE) for its biosynthesis from tryptophan. All the five genes were stacked via yeast assembly and infiltrated into *Nicotiana benthamiana* leaves. Violacein production was observed via LC-MS methods.

tested both cytosolic and plastid expression of bisabolene synthase in tobacco and showed that both constructs were able to produce bisabolene at low levels (Fig. 2b). Next, we stacked a chloroplast-targeted FPP synthase, yielding marginal improvements in yield, an approach previously demonstrated to work[13]. Finally, we stacked the enzyme 1-Deoxy-D-xylulose 5-phosphate synthase (DXS), the first committed step to the plastid-specific non-mevalonate pathway, yielding another five-fold increase in bisabolene production. The combination of assembling various stacked constructs and quickly testing their additive effects through transient expression in tobacco highlights a framework in which one can efficiently assemble, screen and optimize heterologous production of target molecules in plants.

To further demonstrate the capabilities of our DNA assembly system, complex bacterial pathways can be heterologously introduced into plants. We have assembled the entire pathway of violacein from the soil bacterium *Chromobacterium violaceum*. Violacein has been shown to be effective as a therapeutic with both anticancer and antimicrobial properties[14]. The five genes involved in violacein biosynthesis were assembled into pYB vectors and expressed in tobacco leaves. Violacein and various intermediates were detected and confirmed with liquid chromatography mass spectrometry (LC-MS) analysis in infiltrated leaves (Fig. 2c, Supplementary Table 4). The accumulation of the 2-Imino-3-(indol-3-yl)propanoate (IPA imine) intermediate (Supplementary Fig. 2) suggests that the VioB enzyme may be the main rate-limiting step in this violacein pathway. Future efforts may leverage the library of promoters and jStack assembly platform to test various combinations of gene cassettes to improve upon the yields of violacein *in planta*. The ability to easily stitch genes together to reconstruct synthetic metabolic pathways has implications for the future of biofortified foods, production of secondary metabolites and crop resistance to pathogens, to name just a few possible applications.

**Targeted assembly of chromosomal regions from plant genomes.** Flexibility and compatibility are key characteristics for any robust DNA assembly method. For this reason, assembly into pYB vectors can easily be expanded beyond our hierarchical cloning strategy, as it is amenable to a large array of DNA assembly methods as long as there are overlapping, homologous sequence[15]. The plasticity of our method can be highlighted through the ease in assembling large stretches of DNA, such as plant gene clusters or quantitative trait loci. Analogous to bacterial operons, plant gene clusters are chromosomal regions that have evolved to retain multiple genes physically linked near one another involved in specific processes[16]. Many of these clusters are involved in the specific production of secondary metabolites[17] or resistance genes involved in plant pathogen resistance[18]. To demonstrate how jStack can be used to assemble large genetic regions, we assembled a family of nine resistance gene clusters scattered across the *Arabidopsis* genome into nine separate pYB vectors (Supplementary Table 5)—all related to the well-studied Rps4-Rrs1 gene cluster involved in resistance to at least three known pathogens[19]. By simply designing primers with overlapping regions for recombination, all the gene clusters in this family were assembled into pYB vectors (Fig. 3). jStack may enable the assembly of whole libraries of gene clusters from various plant genomes into pYB vectors to easily screen, test and shuffle traits between plant species.

The ease of assembling gene clusters opens the door to the new possibility of designing and constructing synthetic gene clusters. Because each gene cluster confers a specific trait, the stacking of multiple gene clusters would be an efficient means of delivering

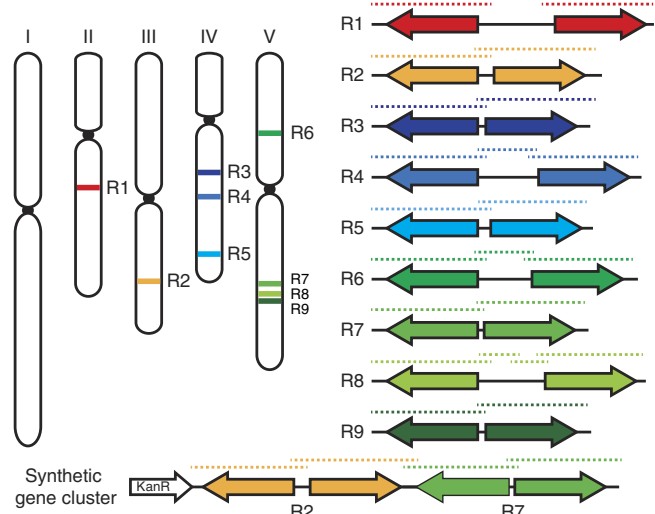

**Figure 3 | High-throughput assembly of R gene clusters and synthetic gene clusters.** Chromosomal positions of all nine R gene clusters (R1–R9) across the *Arabidopsis* genome that display the canonical head-to-head inverted tandem arrangement. Dotted lines represent the PCR amplicons that contain overlapping homologous regions to facilitate yeast homologous recombination into pYB vectors. The number of fragments chosen to assemble a whole gene cluster was dictated by the size of the different gene clusters. Below, assembly of a synthetic gene cluster fusing two R gene clusters together using jStack assembly. Dotted lines over the cartoon schematic of the synthetic gene cluster represent the overlapping PCR amplicons used in jStack assembly of the construct.

multiple traits (for example, varying pathogen resistances) between any plant species. In contrast to conventional breeding methods used to transfer gene clusters or quantitative trait loci between compatible species, the jStack assembly offers a rapid and easy path to minimize the size and manipulate with precision the sequence of the genomic fragments that need to be transferred to one or multiple host plants. Our approach also enables plant scientists to go beyond the breeding compatibility of plant species and replace these constraints with the limit of transformability of recipient species. As a proof-of-concept demonstration, we have built a synthetic gene cluster fusing two of these clusters normally found on separate chromosomes in the *Arabidopsis* genome. We then stably transformed the synthetic gene cluster into a new host, soybean (Fig. 3, Supplementary Fig. 3), showing that three of the four resistance genes were expressed and demonstrating the possibilities of stitching genome-derived gene clusters together for the ultimate purpose of stacking plant resistance traits to various pathogens. Expression of the fourth resistance genes may not have been detected owing to low expression of the genes under our conditions or decreased compatibility of the promoters across species. Future studies leveraging jStack assembly to shuffle gene clusters between species may be most successful between more closely related species, for example, transferring genomic regions from wild relatives to domesticated species.

Altogether, all 10 gene clusters (varying in size between 10.6 and 20.5 kb) were yeast assembled from products of Gibson cloning or direct PCR amplicons, demonstrating the versatility of yeast homologous recombination (Supplementary Fig. 3). Given that entire bacterial genomes have been assembled via yeast homologous recombination, our method provides the opportunity to transfer entire chromosomal regions between species that would not naturally be compatible for breeding. Although there is undoubtedly an upper limit to the amount of DNA that can be

transferred from via *Agrobacterium*-mediated transformation, previous studies have transferred bacterial artificial chromosomes into plants from binary vectors. Liu *et al.*[20] reported only a slight decrease in transformation efficiency by 37% when comparing the transfer of an 80 kb versus a 4.4 kb T-DNA via *Agrobacterium*. Other studies have reported the transfer of T-DNA as large as 150 kb (ref. 21). Although our study does not assemble anything over 20 kb, we envision that most plant engineering efforts will fall well under total size of reported large T-DNAs.

## Discussion

For the ease of sophisticated metabolic engineering, plants are often significantly less tractable to work with than microbes; nonetheless, DNA assembly and cloning should not be a hurdle impeding plant scientists. To address this issue, we have developed a flexible method to streamline large-scale DNA assembly into pYB vectors using a publicly-available library of over 100 DNA parts. Beyond the introduction, manipulation and engineering of plant metabolic pathways, this provides scientists with the tools to redesign, assemble and shuffle whole stretches of synthetic gene clusters to shuffle complex traits into any transformable plant species. Moreover, pYB vectors will also facilitate engineering efforts in other systems, as *Agrobacterium* can transform an array of species including various fungi and human cells[22–24]. The versatility of the jStack platform and its emphasis on establishing open-source resources will be pivotal in establishing a collaborative plant synthetic biology community.

## Methods

**Generation of pYB vectors and parts library.** pYB vectors were built by modifying the pCAMBIA2301 binary vector, conferring kanamycin resistance for bacterial selection (http://www.cambia.org) by incorporating a 391 bp yeast autonomous replicating sequence (ARS4), a 117 bp yeast centromere (CEN6) and a 2,226 bp Leucine selectable marker cassette from pGG092 (a kind gift from Dr John Dueber's lab). Within the left and right borders of the pYB vector, the 722 bp octopine synthase (ocs) terminator (from *Agrobacterium* genomic DNA) and 253 bp nopaline synthase (nos) terminator (from pCAMBIA2301) flanking a 1,103 bp Ura3 cassette (from pDRf1-GW[25]) serve as homology arms for *in vivo* yeast homologous recombination of stacked DNA cassettes. Also within the left and right borders is one of three different plant selectable markers (summarized in Supplementary Table 1) resulting in the three different pYB vectors available: hygromycin (pYB1301; KX817179), kanamycin (pYB2301; KX817180) and Basta (pYB3301; KX817181) resistance. Two intermediate vectors enabling Level 1 assembly of gene cassettes were built: pPMS008 and pPMS028 (Supplementary Table 1). pPMS008 contains a ColE1 ori and thus is a high copy plasmid. pPMS028 is a low–medium copy plasmid due to its p15A ori, allowing researchers to address potential toxicity effects if cloning difficulties arise from cloning into high copy plasmids such as pPMS008. One hundred and fifteen plant-specific DNA parts were cloned with the standard protocol provided with Q5 polymerase (New England Biolabs) for the Level 0 DNA parts library using the corresponding primers described in Supplementary Table 6—a majority of which were subcloned into the backbone pBca9145 (a kind gift from Dr Chris Anderson's lab). All the DNA parts in the library are publicly available through the JBEI ICE registry[9] and are summarized in Supplementary Table 2. All annotated vector maps for Level 0, Level 1 and Level 2 (pYB vectors) are available through the JBEI ICE database. Stacked constructs assembled for this study are further described in Supplementary Table 3.

**jStack assembly process.** For Level 1 assembly, Level 0 parts are digested with their respective Type IIS restriction enzyme (New England Biolabs). Although Level 1 constructs can be assembled via one pot Golden Gate assembly[4], we find that gel extractions (Zymo Research) of restriction enzyme-digested DNA parts allows for mixing and matching of DNA fragments of Level 0 parts that utilize different Type IIS restriction enzymes (for example, BsaI or BsmBI). Moreover, as only a small amount of the gel extracted product is necessary for the subsequent ligation of the Level 1 cassette, we routinely freeze the remaining amount for future use, as we have experienced robust ligation efficiencies with pre-digested DNA parts that have been stored up to half a year at −20 °C. Ligations are transformed into homemade chemically competent *E. coli* DH5α cells and plated on LB plates with chloramphenicol at 37 °C. Plates are screened over a blue light transilluminator (Invitrogen G6500) or an ultraviolet transilluminator for white colonies, as uncut backbone (pPMS008 or pPMS028) will result in green

fluorescent colonies due to the GFP dropout-cassette that should be replaced with correctly assembled Level 1 cassettes.

For Level 2 assembly, Level 1 cassettes are freed from their backbone by choosing one of three eight-base pair rare cutters, NotI, AscI, SbfI (Thermo Fisher Scientific), flanking both ends of assembled DNA cassettes. pYB vectors are digested with the restriction enzyme PmlI (also known as Eco72I, Thermo Fisher Scientific) to linearize the backbone from the Ura3 dropout-cassette, thus exposing both recombination arms (the octopine synthase and nopaline synthase terminator sequences) for subsequent yeast homologous recombination. All plasmids are digested for 2 h at 37 °C and the enzymes are heat inactivated at 80 °C for 5 min. Two microlitres of each digested fragment (~50–100 ng) and 2 μl of the digested pYB vector (~50–100 ng) are transformed into yeast (strain BY4742; MATα his3Δ1 leu2Δ0 lys2Δ0 ura3Δ0) using the Zymogen Frozen-EZ Yeast Transformation II kit. There is no need to clean up or further purify digested Level 1 fragments after heat inactivation. Transformed yeast are plated on solid Synthetic Complete Yeast Media—Leu + 2% glucose + 5-fluoroorotic acid (1 g l−1; Zymo Research, F9001-5) providing negative selection of uncut pYB vectors that have retained the Ura3 dropout-cassette. DNA assemblies in the pYB vector are recovered from yeast (Zymoprep II kit, Zymo Research) and electroporated into *E. coli* (ElectroMAX DH10B, Thermo Fisher Scientific) for plasmid amplification and sequence verification.

To assess the efficiency of the jStack assembly, we assembled the largest construct described in this study, the synthetic gene cluster, which simultaneously assembled four DNA fragments into a pYB vector to produce a plasmid over 32 kb in size. From 24 yeast colonies, 14 were sequenced verified to have the correctly assembled synthetic gene cluster—a success rate of 58%. These rates of efficiency for DNA assembly enable researchers to routinely screen two yeast colonies with a good chance to recover fully assembled and stacked DNA construct.

**Production of bisabolene in tobacco leaves.** *Nicotiana benthamiana* plants were grown in growth chambers (Percival-Scientific) at 25 °C in 16/8 h light/dark cycles with 60% humidity. Leaves of 4-week-old plants were infiltrated with *Agrobacterium tumefaciens* strain GV3101 (OD600 = 1.0) carrying pYB vectors of interest following the procedure described in Sparkes *et al.*[26]. Various genes were stacked into one plasmid for the production of bisabolene from one strain of *Agrobacterium*. A Nopaline Synthase promoter was used to drive Bisabolene Synthase (or its corresponding gene harbouring a chloroplast localization sequence). The MAS (Mannopine Synthase) promoter was used to express the FPP Synthase. The RbcS (Rubisco Small Subunit) promoter was used to drive the DXS gene (summarized in Supplementary Table 3). Infiltrated plants were returned to the same growth conditions after infiltration. Roughly 100–200 mg of fresh leaf tissue was collected 4 days post infiltration, weighed out and stored at −80 °C until further analysis. The leaf samples were frozen in liquid nitrogen and ground with a bead-beater (Qiagen, TissueLyser) at maximum speed for 5 min. One millilitre of a 85:15 solution of hexane:ethyl acetate was added to each sample and homogenized in the bead-beater for 15 min. The samples were transferred into a 2 ml microtube and spun down at maximum speeds for 5 min at room temperature to clear off the supernatant that was then taken for further analysis. Bisabolene was quantified in the supernatant by gas chromatography mass spectrometry (Thermo Trace Ultra with PolarisQ MS) as previously described[10] with a TR-5MS column (30 m × 0.25 mm ID × 0.25 μm film) using a bisabolene standard curve using the following conditions: inlet at 250 °C, 1.1 ml min−1 constant flow, transfer line at 250 °C, ion source at 230 °C, scan *m/z* 50–300. Oven: 100 °C for 0.75 min, ramp at 40 °C min−1 to 300 °C, hold 2 min.

**Production of violacein in leaves.** A pYB plasmid was assembled with all five genes necessary to produce violacein: the Hsp18, MAS, Actin2, RbcS and Bch1 promoters were used to drive expression of VioA, VioB, VioE, VioC and VioD, respectively (summarized in Supplementary Table 3). *N. benthamiana* leaves were infiltrated with *Agrobacterium*, as described above. For each sample, three leaves on one *N. benthamiana* plant were harvested 5 days post infection. Leaves were weighed out, then snap frozen in liquid nitrogen, and ground with mortar and pestle. One millilitre of 100% MeOH was added to ground material for the extraction of soluble metabolites. Each sample was heated at 80 °C and shaken for 15 min. Each sample was then spun down and the MeOH supernatant was kept, and another 1 ml of 100% MeOH was added to resuspend and wash the pellet. This step was repeated five times and for each samples, all the collected MeOH fractions were pooled together before further analysis.

In preparation for LC-MS, each MeOH extract was dried in a SpeedVac (Thermo Savant SPD111) at room temperature, then re-suspended in 600 μl to 1.5 ml MeOH containing an internal standard (2-amino-3-bromo-5-methylbenzoic acid, 1 μg ml−1, Sigma, #631531) and homogenized using a water bath sonicator for 1 h at room temperature. The samples were centrifuged at 3,220*g* for 5 min, supernatant removed, then 200 μl of supernatant centrifuge-filtered through a 0.22 μm polyvinylidene fluoride membrane (UFC40GV0S, Millipore). LC-MS/MS (liquid chromatography tandem mass spectrometry) was performed on a 2 μl injection, with UHPLC reverse-phase chromatography performed using an Agilent 1290 LC stack and Agilent C18 column (ZORBAX Eclipse Plus C18, Rapid Resolution HD, 2.1 × 50 mm, 1.8 μm) and with MS and MS/MS data collected using a QExactive Orbitrap mass spectrometer (Thermo Scientific, San Jose, CA,

USA). Chromatography used a flow rate of 0.4 ml min$^{-1}$, first equilibrating the column with 100% buffer A (LC-MS water with 0.1% formic acid) for 1.5 min, then diluting over 7 min to 0% buffer A with buffer B (100% acetonitrile with 0.1% formic acid). Full MS spectra was collected at 70,000 resolution from $m/z$ 70–1,050 and MS/MS fragmentation data collected at 17,500 resolution using 10, 20 and 30 eV collision energies (averaged together). Violacein was identified in samples based on accurate mass $m/z$, and comparing retention time and fragmentation spectra to a purchased standard (violacein from *Janthinobacterium lividum*, V9389, Sigma). Identification of violacein biosynthesis pathway intermediates and shunt products was based on accurate mass $m/z$. Detected mass and retention time peak for each detected compound are shown in Supplementary Table 4, as well as relative intensity for each peak (Supplementary Fig. 2).

**Generation of stably transformed soybean roots.** For induction of soybean hairy roots, cotyledons were inoculated as previously described[27] with some modifications. Transformed *Agrobacterium rhizogenes* strain NCPPB2659 (K599 kindly provided by S. Senthil, South Dakota State University) harbouring the described pYB vectors was grown in 5 ml liquid LB broth with antibiotics (Kanamycin, 100 mg ml$^{-1}$) at 28 °C on a rotary shaker at 200 r.p.m. overnight. The overnight culture was then used to inoculate 100 ml LB broth containing the same antibiotics (Kanamycin, 100 mg ml$^{-1}$) that was then incubated at 28 °C on a rotary shaker at 200 r.p.m. overnight. The cells were collected by centrifugation at 3,000*g* for 15 min, and resuspended in MS-N basal medium (pH 5.8; PhytoTechnology Laboratories, M531) to an OD600 of 0.4–0.5. Sterilized Fibrgro cubes (1 cm$^3$; Hummert International, Earth City, MO, USA) were saturated with resuspended cell cultures. The apical stem sections of young plants, collected during vegetative growth phase and grown at 22 °C in 14/10 h light/dark cycles were inserted into the inoculated Fibrgro cubes within plant growth trays and covered with plastic domes and incubated at 25 °C with a 16/8 h light/dark cycles under an illumination of 40 μmol m$^{-2}$ s$^{-1}$. The cube moisture was checked every 3–4 days and watered with ½ MS –N media (PhytoTechnology Laboratories) when necessary for the remainder of the induction period until roots emerged from the teratoma. After the formation of hairy roots, the composite plant roots were subsequently analysed by microscopy or RT–PCR (PCR with reverse transcription) analysis.

**Microscope analysis of transformed roots.** A laser scanning confocal microscope (LSM 710; Carl Zeiss Microscopy) was used for fluorescence analysis of *Glycine max* hairy roots stably transformed with the three reporter genes. Excitation of GFP and DsRed was performed using lasers at 488 with emission filter 510–530 nm and 558 nm with emission filter 583–592 nm, respectively Four-week-old hairy roots from soybean expressing GFP and DsRed were used for imaging.

**Histochemical GUS staining.** To assay β-glucuronidase (GUS) reporter activity, whole DsRED- or GFP-positive roots were infiltrated with staining solution (1 mM EDTA, 0.2% Triton X-100, 0.2% Tween-20 in TBS, pH 7.3) containing 1 mM 5-bromo-4-chloro-3-indolyl-β-D-glucuronide (X-Gluc). Ferricyanide (0.25 mM) was added to prevent indigo precursor migration[28]. The chelator EDTA was added to the staining solution to prevent any gene expression during the staining procedure. Substrate penetration was assisted by 5 min sonication and subsequently two vacuum infiltrations at 0.1 atm for 5 min each to improve infiltration. The roots were incubated in staining solution at 37 °C until sufficient blue staining had been developed.

**Gene cluster DNA assembly.** All bidirectional gene clusters in the Rps-Rrs1 family encoded in the *Arabidopsis thaliana* genome were previously identified[19] and summarized in Supplementary Table 5. Primers were designed for PCR amplicons to cover the entirety of all gene clusters, with each amplicon ranging between 5 and 9 kb in size. The number of fragments chosen to assemble a whole gene cluster was dictated by the size of the different gene clusters. If a given amplicon did not amplify on the first attempt, the amplicon was divided into two halves containing an overlapping sequence, increasing the chance of successful PCR owing to the smaller amplicon size. The primers were also designed to allow at least 200 base pairs of overlap between amplicons to allow for *in vivo* yeast homologous recombination and reconstitution of gene cluster in the pYB vector. All gene clusters were broken into either two or three PCR amplicons (Supplementary Table 7). Linker (L_tOcs; primers PMS1465/PMS1466) and terminator (T_tNos; primers PMS1566/PMS1567) sequences were adapted onto flanking amplicons using Gibson cloning (New England Biolabs, #E22611S) to enable recombination with the pYB vector backbone. The primers are summarized in Supplementary Table 7. The PCR amplicons were gel extracted (Zymo Research, #D4002) and transformed in yeast as described above with the linearized pYB vector, pYB2301. The synthetic gene cluster fusing clusters R2 and R7 were assembled by adapting the 3′ end of the R2 cluster to the 5′ end of the R7 cluster via Gibson assembly (summarized in Supplementary Fig. 3). All gene clusters were assembled into the pYB vector pYB2301.

**RT–PCR of the resistance gene cluster.** RNA was isolated using the RNEasy plant mini kit (Qiagen, http://www.qiagen.com) from hairy roots of 4-week-old *Glycine max* composite plants. First-strand cDNA synthesis and subsequent PCR

were carried out using 1 μg of total RNA according to the Tetro cDNA synthesis protocol (Bioline). One microlitre of RT reaction were used as template in 50 μl final reaction volume for amplicon quantification after PCR using gene-specific primers (Supplementary Table 7) and DreamTaq Green DNA Polymerase (ThermoScientific) following the manufacturer's recommendations. Ubiquitin (GLy20g27950) was used as the endogenous control with primers: F: 5′-ACCCT TCACCTTGTCCTCCGTC-3′, and R: 5′-GACACATTGAGTTCAACACAAAC CG-3′. Expression of transgenes from the synthetic gene cluster was assayed for neomycin phosphotransferase II (primers: F: 5′-GGAGAGGCTATTCGGCTA TG-3′, and R: 5′-GCCAACGCTATGTCCTGATA-3′); At5g45060 (primers: F: 5′-ATGTTGCAAAACCACTGGCC-3′, and R: 5′-TCCAACCGCTTCATGACC AA-3′); At5g45050 (primers: F: 5′-GTCTCTCATACGTGTATCTTCCAATGG-3′, and R: 5′-AAACATTTGGATCAGTGCGG-3′); and At3g51560 (primers: F: 5′-AT TGCATTCCCCAAAGTCAA-3′, and R: 5′-TTGCGACCTATGGATTGGAT-3′).

**Data availability.** The authors declare that the data supporting the findings of this study are available within the article and its Supplementary Information Files are available from the corresponding author upon request. All vectors and resources described are publicly available and can be found through the Inventory of Composable Elements (ICE) at https://acs-registry.jbei.org/. Once logged in, the plasmids and strains are listed under the JBEI Public Registry tab. Finally, the sequences of pYB1301 (KX817179), pYB2301 (KX817180) and pYB3301 (KX817181) were deposited to the NCBI nucleotide database.

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

## Acknowledgements

We thank James Kirby and Ee-Been Goh for technical expertise and helpful comments on bisabolene experiments. This work was part of the DOE Early Career Award and the DOE Joint BioEnergy Institute (http://www.jbei.org) supported by the U.S. Department of Energy, Office of Science, Office of Biological and Environmental Research; and U.S. Department of Energy Joint Genome Institute, a DOE Office of Science User Facility through contract DE-AC02-05CH11231 between Lawrence Berkeley National Laboratory and the U.S. Department of Energy. The United States Government retains and the publisher, by accepting the article for publication, acknowledges that the United States Government retains a non-exclusive, paid-up, irrevocable, world-wide license to publish or reproduce the published form of this manuscript, or allow others to do so, for United States Government purposes. This project has received funding from the European Union's Horizon 2020 research and innovation programme under the Marie Sklodowska-Curie grant agreement N° 659910. P.M.S. was supported by the Gordon and Betty Moore Foundation through Grant GBMF 2550.04 to the Life Sciences Research Foundation.

## Author contributions

P.M.S. and D.L. designed the project. P.M.S., N.M., K.V., L.A., K.B.L. and B.P.B. performed the experiments and analysis. K.B.L., B.P.B. and T.R.N. performed the violacein analyses. P.M.S. wrote the paper. D.L. and T.R.N. edited the document and provided financial support and supervision.

## Additional information

**Competing financial interests:** D.L. has financial interests in Afingen. The remaining authors declare no competing financial interests.

