## [Peer Review File · Nature Communications]

Reviewers' comments:

Reviewer #1 (Remarks to the Author):

The authors report a new strategy, called jSTACK, for assembly of multi-locus plasmids for use in plant transformation, as well as a library of more than 100 standardized plant parts (e.g., promoters, linkers, terminators). The authors have identified a major technological gap in plant synthetic biology: the ability to rapidly introduce multiple transgenes into plants. *Agrobacterium* inserts DNA at random sites, leading to highly variable expression levels among independent transformants, and, in addition, transformation rates are modest (standard methods for stable transformation yield ~0.1-1% efficiency and much lower efficiency is observed for multiple insertion events from mixed *Agrobacterium* inoculations). These factors create a critical bottleneck for applications like metabolic engineering where multiple new proteins need to be expressed at relatively fixed stoichiometric ratios. The jSTACK method solves several problems at once and is completely open-source and cross-compatible with most existing DNA-stitching technologies. I have no doubt that many labs all over the globe will adopt it as soon as it is made public. Overall, the manuscript is very well written, and the figures are clear and easy to follow. Data support conclusions and appear to be analyzed appropriately.

Suggested improvements:

-The use of trinary to describe the new destination vectors should be reconsidered. Binary vectors were so-named because they split the function of the genes on the Ti plasmid into two plasmids—a small plasmid containing the T-DNA that could be readily manipulated with standard molecular cloning methods and a larger 'helper' plasmid that carried all of the vir genes needed for transfer of the T-DNA into plants. The new vectors used here still rely on a helper plasmid and, thus, are still part of a binary transformation system. The use of trinary implies that a third type of plasmid is required for transformation, which is not the case.

-Please explain the origin of the jSTACK name.

-The jSTACK method is clever and potentially quite powerful. The impact would be greater if the authors provided quantitative benchmarks. For example, how is transformation efficiency affected by size of T-DNA? It would be great to see a simple comparison of transient transformation efficiency with each additional marker added to the T-DNA. In addition, the authors argument would be stronger if they provided a comparison of the efficiency of dual transformation when using a mixture of two *Agro* strains each carrying a single marker versus the efficiency of using a single *Agro* strain made with jSTACK. Also, the authors should include whatever is known about the maximum size of an insert, both for amplification in *Agro* and for efficient insertion into the plant genome (and the associated references).

-Among the large number of plant parts in the library presented here, how many were experimentally validated? It would be nice to know how many of the promoters, terminators, linkers worked in these researchers hands. Were any of them easier/harder to work with in PCR amplification or assembly? If only a smaller core set were examined in

detail, perhaps these could be separated out as the validated set and annotated with this type of practical detail.

- The sequence of all primers used for assembly of any of the plasmids should be included.

-A minor note: Homologous recombination should be spelled out throughout the text to improve readability.

Reviewer #2 (Remarks to the Author):

"A robust gene stacking method utilizing yeast assembly for plant synthetic biology"

In this manuscript, Shih et al. describe the development of a method to efficiently clone several genes (with their respective transcriptional promoters and terminators) intended for expression in plants or plant cells. This method is named jSTACK, and it consists of a three-tiered assembly, where Level 0 plasmids are initially assembled via standard molecular biology techniques. The authors built a library of over one hundred Level 0 plasmids containing publicly available plant-specific parts. These plasmids are then used for assembling Level 1 plasmids using Type IIS restriction enzymes (Golden Gate assembly). Finally, "trinary" plasmids that can be maintained in *E. coli*, *Agrobacterium* and yeast are assembled by using homologous recombination in yeast cells.

The authors initially demonstrate the method by stacking three reporter genes and transforming soybean roots. Next, they assemble a few plasmids with increasing number of genes involved in biosynthesis of the chemical bisabolene, followed by assembly of a plasmid containing the entire pathway for violacein biosynthesis. Finally, large plant gene clusters are assembled in these trinary plasmids for plant transformation and expression.

This is an interesting and creative method that takes advantage of homologous recombination in yeast to assemble the final plant transformation plasmids. The availability of 115 Level 0 plasmids containing publicly available plant parts will be very useful to the plant research community. The jSTACK method has the potential to significantly enhance plant molecular biology and synthetic biology by improving the efficiency of cloning and gene stacking for plant transformation. However, in its present form, the manuscript is lacking some basic information, as detailed below, which makes it difficult to judge how likely it is to be adopted by the community, and therefore at this time I cannot recommend publication in Nature Communications.

The main issue with this manuscript is that it is not clear how jSTACK compares in terms of efficiency to other methods currently available for assembling plant transformation methods, such as using Golden Gate for all levels of assembly (Binder et al., PLoS ONE 9 (2): e88218, 2014; Engler et al., ACS Synth. Biol. 3: 839-843, 2014), Gibson assembly and Gateway technology (Karimi et al., Trends in Plant Sci. 7 (5): 193-195, 2002; Dalal et al., Plasmid 81: 55-62, 2015). The manuscript is missing some metrics, such as for example how many yeast clones were obtained, and what percentage contained the correct inserts.

Without metrics, it is difficult to judge how much of an improvement this method provides to what is currently done in the field. If the method does not provide an advantage over current methods, it is unlikely to be adopted by the research community, especially as it introduces a third organism, yeast, that needs to be cultured, in addition to *E. coli* and *Agrobacterium*. If the main point of the paper is to describe the new method, as the title implies, then a comparison between stacking genes with jSTACK and a conventional method is warranted.

Additionally, the authors claim that they have built a synthetic gene cluster fusing two clusters (R2 and R7) normally found on separate *Arabidopsis* chromosomes. However, there is no experimental evidence that all four genes in these two clusters were successfully cloned into the plant transformation plasmid. Further, RT-PCR results only show one of these genes being expressed in soybeans. Are the other three genes not expressing? Why not? Are they even in the plasmid?

Again, given that the main goal of the paper is to describe a new plasmid assembly method, the lack of evidence that the plasmid is correctly assembled prevents one from judging whether this constitutes an improvement over currently available method and makes it unlikely to be adopted widely by the plant research community.

Minor issues:

Lines 148-149 - "We targeted the molecule... This sentence is not clear.

Lines 153-154 - "First, we tested both cytosolic and plastid expression of bisabolene synthase in tobacco". Where are the results for the individual expression of BisSyn in chloroplasts? This is not shown.

Line 186 - "...we assembled a family of nine R gene clusters..." This sentence is misleading, as it seems to indicate that all nine gene clusters were assembled into one plasmid, and this is not shown in the results.

Lines 241, 243 - NCBI numbers are missing.

Line 264 - DH5alpha.

Line 311 - "...in order for extracted." This sentence does not make sense.

Line 314 - "...each sample..." (not samples); "...together prior to further analysis."

Line 358 - "...analyzed by microscopy..."

Line 514 - "...ends are denoted..."

Line 517 - "Compatibility of liberated..."

Line 533 - "...GFP and DsRed fluorescence."

Line 541 - abbreviation for DXP is not defined.

Figure 3a - it would be helpful to explain the criteria for determining the extent of overlap between fragments. Some assemblies have 2 overlapping sequences, while others have 3 or 4. Why is this?

Reviewer #3 (Remarks to the Author):

The manuscript by Shih et al. describes a new vector system designed for rapid assembly of constructs for heterologous expression in plants. The authors are correct that tools for plant synthetic biology are lacking, and overall, I believe that their system is novel and has the potential to be exceptionally useful. I have some comments and questions that I think should be addressed before publication.

1. The authors describe in detail the overall construction of the vector and the basic strategy of how genes are placed in it. However, while the authors provide an impressive list of 115 parts in Supp. Table 2, very little is said about the compositions/applications of these parts. I believe that this information is provided in the JBEI website, though I don't have an account to this website so I haven't checked personally. Could the authors explicitly describe where information about the 115 parts can be found, providing the website in the text.

2. Some details in the results section regarding the three engineering examples provided would also give a better perspective of the wide range of tools available. In the reporter stacking, bisabolene and violacein production, and resistance gene examples, I did not see what promoter(s) were used in the main text or Methods. I was also not clear from the text if stacking always meant stacking of genes in the vector, or stacking of multiple vectors, each containing a single gene. For example, for violacein, were five agro strains, each harboring one gene, or one construct harboring all five genes? These details are important and would provide a much clearer picture of the system.

3. With the violacein pathway, the supplementary data shows that a couple of biosynthetic intermediates accumulate in addition to the final product. While it is beyond the scope of this manuscript to optimize this pathway, some minimal discussion of the results of the engineered pathway - such as what intermediates accumulate, and possibly why they do - and also some mention of how the power of this expression system could be used to optimize the production would be valuable.

4. An exciting part of the manuscript describes the cloning of plant genomic regions, or clusters. The authors showed that two R gene clusters could be stitched together and transformed into soybean hairy roots. The authors state that at least one of the R genes was expressed as shown in Figure 3. Why was the expression of only one gene tested, when, as I understand the figure, 4 R genes were placed into the vector?

5. Some discussion of the limits of the insert size would also be helpful, especially for the non-specialist. Notably, the gene clusters that encode secondary metabolites are much

larger than the 10-20 kbp used for the R genes.

Reviewer #4 (Remarks to the Author):

In the manuscript of Shih et al., the authors present a method called jSTACK to assemble multiple DNA fragments making use of yeast homologous recombination (HR). They tailored a regular plant binary vector into a so-called trinary vector for extra plasmid propagation in yeast, and designed a two-tier strategy to streamline DNA assembly. They demonstrated a few examples of using jSTACK to stack 2-5 gene expression cassettes into a trinary vector and expressing these genes in *Nicotiana benthamiana* leaves or soybean roots via *Agrobacterium*-mediated transformation. The jSTACK method is undoubtedly a good complement to current DNA assembly tools for plant synthetic biology research.

Major concerns:

1. The advantage of jSTACK over the Gibson assembly (GA) has not been addressed. Although yeast HR can facilitate multiple DNA assembly, the jSTACK method requires yeast and a customized trinary vector as additional experimental materials that are not readily available in plant research laboratories. In contrast, GA can be conducted with homemade or commercially available reagents in a cell-free system.
2. The authors stacked at most 5 genes in this work, which is not impressive at all, and can routinely be done with the Golden Gate Assembly (GGA) or GA. The authors need to show some cases where gene stacking can be easily accomplished with jSTACK but can barely be done with GGA or GA.
3. The prerequisite for the ultimate usefulness of the jSTACK method is that multiple genes stacked by jSTACK can be stably transformed and successfully expressed. Therefore, data of stable transgenic expression of multiple stacked genes in WHOLE plants (not only in stably transformed soybean roots) are necessary. Moreover, evidence of successful expression of individual stacked genes (e.g., R genes in Fig.3) needs to be provided.

Response to reviewers:

Reviewer #1 (Remarks to the Author):

The authors report a new strategy, called jSTACK, for assembly of multi-locus plasmids for use in plant transformation, as well as a library of more than 100 standardized plant parts (e.g., promoters, linkers, terminators). The authors have identified a major technological gap in plant synthetic biology: the ability to rapidly introduce multiple transgenes into plants. *Agrobacterium* inserts DNA at random sites, leading to highly variable expression levels among independent transformants, and, in addition, transformation rates are modest (standard methods for stable transformation yield ~0.1-1% efficiency and much lower efficiency is observed for multiple insertion events from mixed *Agrobacterium* inoculations). These factors create a critical bottleneck for applications like metabolic engineering where multiple new proteins need to be expressed at relatively fixed stoichiometric ratios. The jSTACK method solves several problems at once and is completely open-source and cross-compatible with most existing DNA-stitching technologies. I have no doubt that many labs all over the globe will adopt it as soon as it is made public. Overall, the manuscript is very well written, and the figures are clear and easy to follow. Data support conclusions and appear to be analyzed appropriately.

Suggested improvements:

-The use of trinary to describe the new destination vectors should be reconsidered. Binary vectors were so-named because they split the function of the genes on the Ti plasmid into two plasmids—a small plasmid containing the T-DNA that could be readily manipulated with standard molecular cloning methods and a larger 'helper' plasmid that carried all of the vir genes needed for transfer of the T-DNA into plants. The new vectors used here still rely on a helper plasmid and, thus, are still part of a binary transformation system. The use of trinary implies that a third type of plasmid is required for transformation, which is not the case.

> *It was not our intent to confuse the name with the split binary vector system. Accordingly, we have taken the Reviewer's concern into consideration and renamed the use of yeast vectors to Yeast-compatible Binary vectors, or pYB vectors.*

-Please explain the origin of the jSTACK name.

> *We have added the following text and omitted the capital letters to avoid any confusion, as the name "jSTACK" is not an acronym.*

"Given the intent of this DNA assembly platform to stack DNA and genes and the institution in which it was developed (Joint BioEnergy Institute) we have correspondingly named our jStack."

-The jSTACK method is clever and potentially quite powerful. The impact would be greater if the authors provided quantitative benchmarks. For example, how is transformation efficiency affected by size of T-DNA? It would be great to see a simple comparison of transient transformation efficiency with each additional marker added to the T-DNA. In addition, the authors argument would be stronger if they provided a comparison of the efficiency of dual transformation when using a mixture of two *Agro* strains each carrying a single marker versus the efficiency of using a single *Agro* strain made with jSTACK. Also,

the authors should include whatever is known about the maximum size of an insert, both for amplification in Agro and for efficient insertion into the plant genome (and the associated references).

> The transformation efficiency of large T-DNAs has already been examined in the literature. Liu et al, PNAS, 1999 compared the efficiency between a 80kb T-DNA and a 4.4kb T-DNA, observing only a slight decrease in efficiency where the 80kb T-DNA transformation occurred at a rate 63% of the smaller 4.4kb T-DNA. Although we find that most plant molecular labs will be fine keeping T-DNAs under 80kb for the vast majority of their projects, we believe that it may be worthwhile in the future to follow-up on this study by developing derivative pYB plasmids or new agrobacterium strains that may increase the efficiency of transfer of larger than 80kb T-DNAs into plants.

We have included the following text with associated references for the maximum size of inserts/T-DNAs:

“Although there is undoubtedly an upper limit to the amount of DNA that can be transferred from via Agrobacterium mediated transformation, previous studies have transferred bacterial artificial chromosomes (BACs) into plants from binary vectors. Liu et al. reported only a slight decrease in transformation efficiency by 37% when comparing the transfer of an 80 kb versus a 4.4 kb T-DNA via Agrobacterium²⁰. Other studies have reported the transfer of T-DNA as large as 150 kb²¹. Although our study does not assemble anything over 20kb, we envision that most plant engineering efforts will fall well under total size of reported large T-DNAs.”

Furthermore, it is important to point out that the intention of this technology is not to be solely used for transient expression assays and test metabolic pathways; some of the experiments in our study use transient expression assays and stable transformation as a demonstration that the modified pYB vectors could transfer T-DNA into plants. However, pYB vectors can clearly be used to stack multiple genes into one T-DNA, facilitating transient expression experiments of multi-gene metabolic pathways and avoiding the need to infiltrate individual Agrobacteria strains hosting single genes. For example, Crocoll et al (Front Bioeng Biotech 2016) found that expressing the genes involve in the biosynthesis of dihomomethionine (DHM) stacked together produced more than three times higher levels of DHM than when combining individual strains of Agrobacterium, each with only one gene in the pathway. Stable root transformants in soybean and tobacco were made to reiterate the fact that T-DNAs derived from pYB could be stably integrated in the genome of host plants. It also showed that the large T-DNA resulting from the synthetic gene cluster assembly in this study could be co-transferred with the 15 kb T-DNA from the pRi2659 plasmid of the Agrobacterium K599 strain that is required to regenerate roots.

Another point to highlight is that we are not trying to solve the transformation efficiency problem. We are trying to present a novel means to assemble DNA into plant transformation vectors. Even if there is a decrease in transformation efficiency, the ability to stack five genes simultaneously in one T-DNA will still be much faster than transforming one gene at a time successively into one plant host - a common practice that is still done by many plant molecular labs.

-Among the large number of plant parts in the library presented here, how many were experimentally validated? It would be nice to know how many of the promoters, terminators, linkers worked in these researchers hands. Were any of them easier/harder to work with in PCR amplification or assembly? If only a smaller core set were examined in detail, perhaps these could be separated out as the validated set and annotated with this type of practical detail.

> *The DNA parts we chose to add into the library have been previously characterized or described in other studies. We have updated Supplementary Table 2 to add another column that cites the reference that describes a study that has characterized the part. Characterizations are primarily for promoter sequences; however, respective terminators were cloned out, as many times it is common practice to pair a promoter and terminator together from the same gene. In these cases, if a terminator was not explicitly characterized in a study, we have cited the study that describes the promoter characterization, thus explaining the rationale for cloning out the corresponding terminator. Although the majority of the parts have been characterized in the literature for functionality (primarily promoters), our own experimental validation and characterization is limited to the constructs generated for this study specifically. A set of promoters in the library has been characterized by our group in prior publications, and those have been referenced in the new Supplementary Table 2. The vast majority of parts (>95%) of genes were PCR amplified out on the first try, and all amplified by the second attempt. We have added to the Methods describing the polymerase and protocol used.*

- The sequence of all primers used for assembly of any of the plasmids should be included.

> *Supplementary Table 6 has all the primer sequences used to assemble the plasmid library*

-A minor note: Homologous recombination should be spelled out throughout the text to improve readability.

> *To improve readability, HR has been changed to 'homologous recombination' throughout the text*

Reviewer #2 (Remarks to the Author):

"A robust gene stacking method utilizing yeast assembly for plant synthetic biology"

In this manuscript, Shih et al. describe the development of a method to efficiently clone several genes (with their respective transcriptional promoters and terminators) intended for expression in plants or plant cells. This method is named jSTACK, and it consists of a three-tiered assembly, where Level 0 plasmids are initially assembled via standard molecular biology techniques. The authors built a library of over one hundred Level 0 plasmids containing publicly available plant-specific parts. These plasmids are then used for assembling Level 1 plasmids using Type IIS restriction enzymes (Golden Gate assembly). Finally, "trinary" plasmids that can be maintained in E. coli, Agrobacterium and yeast are assembled by using homologous recombination in yeast cells.

The authors initially demonstrate the method by stacking three reporter genes and transforming soybean roots. Next, they assemble a few plasmids with increasing number of genes involved in biosynthesis of the chemical bisabolene, followed by assembly of a plasmid containing the entire pathway for violacein biosynthesis. Finally, large plant gene clusters are assembled in these trinary plasmids for plant transformation and expression.

This is an interesting and creative method that takes advantage of homologous recombination in yeast to assemble the final plant transformation plasmids. The availability of 115 Level 0 plasmids containing publicly available plant parts will be very useful to the plant research community. The jSTACK method has the potential to significantly enhance plant molecular biology and synthetic biology by improving the efficiency of cloning and

gene stacking for plant transformation. However, in its present form, the manuscript is lacking some basic information, as detailed below, which makes it difficult to judge how likely it is to be adopted by the community, and therefore at this time I cannot recommend publication in Nature Communications.

The main issue with this manuscript is that it is not clear how jSTACK compares in terms of efficiency to other methods currently available for assembling plant transformation methods, such as using Golden Gate for all levels of assembly (Binder et al., PLoS ONE 9 (2): e88218, 2014; Engler et al., ACS Synth. Biol. 3: 839-843, 2014), Gibson assembly and Gateway technology (Karimi et al., Trends in Plant Sci. 7 (5): 193-195, 2002; Dalal et al., Plasmid 81: 55-62, 2015). The manuscript is missing some metrics, such as for example how many yeast clones were obtained, and what percentage contained the correct inserts. Without metrics, it is difficult to judge how much of an improvement this method provides to what is currently done in the field. If the method does not provide an advantage over current methods, it is unlikely to be adopted by the research community, especially as it introduces a third organism, yeast, that needs to be cultured, in addition to E. coli and Agrobacterium. If the main point of the paper is to describe the new method, as the title implies, then a comparison between stacking genes with jSTACK and a conventional method is warranted.

> We apologize for the lack of metrics and have added them to the manuscript. We have done so for the largest construct assembled in this study, the synthetic gene cluster. We believe that our transformation efficiency demonstrates the robust nature of yeast assembly, as more than 50% of yeast colonies were correctly assembled (on average one can pick two yeast colonies and reasonably assume that one is correct). Although other methods such as Golden Gate cloning have described assembly efficiencies that are higher, we believe that in the lab, the efficiency of jStack assembly is more than sufficient for large-scale DNA assembly, as more than 50% yeast colonies were correctly assembled, thus on average one can pick two yeast colonies and reasonably assume that one is correct.

We have added the following text to report the percentage of assembly:

“To assess the efficiency of the jStack assembly, we assembled the largest construct described in this study, the synthetic gene cluster, which simultaneously assembled four DNA fragments into a pYB vector to produce a plasmid over 32 kb in size. From twenty-four yeast colonies, fourteen were sequenced verified to have the correctly assembled synthetic gene cluster – a success rate of 58%. These rates of efficiency for DNA assembly enable researchers to routinely screen two yeast colonies with a good chance to recover fully assembled and stacked DNA construct.”

Another point to take into consideration is that the research community will most likely adopt multiple methods, depending on the lab, the scientist and the research project. Thus, we have made our parts library compatible with other Golden Gate systems, but scientists may be more inclined to use jStack because of the advantages in assembling large fragments purely based on homology. For example, jStack enables the assembly of large fragments of DNA that might not be able to realistically be assembled with either Golden Gate or Gateway technology, such as the assembly of large gene clusters. If one were to use the described Golden Gate methods to assemble the same gene clusters, one would have to devoid these sequences of all the BsaI, BpiI, and BsmBI cut sites; within the 10 assembled clusters there were 36 BsaI cut sites, 85 BpiI cut sites, and 30 BsmBI cut sites. Moreover, both Golden Gate and Gateway methodologies introduce ‘scars’ into the assembled sequences. Golden Gate methods introduce the four base pairs necessary for ligation as a scar, whereas Gateway technology introduces larger scars from the

att recombination sites. In addition, the second level of DNA assembly (based on yeast homologous recombination) can be easily made compatible to assemble DNA fragments derived from various cloning methods (Gibson, Golden Gate, Gateway technology, conventional ligation....) including PCR products. Thus, we believe that the jStack assembly platform provides clear advantages over some strategies and will be adopted by the community.

Additionally, the authors claim that they have built a synthetic gene cluster fusing two clusters (R2 and R7) normally found on separate Arabidopsis chromosomes. However, there is no experimental evidence that all four genes in these two clusters were successfully cloned into the plant transformation plasmid. Further, RT-PCR results only show one of these genes being expressed in soybeans. Are the other three genes not expressing? Why not? Are they even in the plasmid?

Again, given that the main goal of the paper is to describe a new plasmid assembly method, the lack of evidence that the plasmid is correctly assembled prevents one from judging whether this constitutes an improvement over currently available method and makes it unlikely to be adopted widely by the plant research community.

> We apologize for the lack of evidence that the genes have been successfully assembled. We have included this data in Supplementary Figure 3b, showing PCR confirmation the presence of the four R genes from the two fused gene clusters assembled into the transformation plasmid. The main purpose of this study was to demonstrate the different capabilities that could be leveraged using a yeast assembly method for plant biotechnology to facilitate DNA assembly. Thus, our focus is not on the expression of the genes from the synthetic cluster, but rather the assembly of the synthetic cluster. Nonetheless, we initially provided RT-PCR data for only one gene to show as a simple proof-of-concept that the gene cluster could be expressed in other plant species.

In order to address the Reviewer's concern, we have attempted to detect expression of all the genes in the synthetic gene cluster. After numerous attempts, we can demonstrate expression of three of the four genes from the synthetic gene cluster and the plant maker gene. We have included this data in Supplementary Figure 3. We encountered difficulty troubleshooting the RT-PCR for At3g51570 for a number of possible reasons: 1) this R gene is expressed at very low level, 2) it is not expressed under this growing conditions, 3) it is repressed by the Agrobacterium genes present in the RiT-DNA, 4) the Arabidopsis R gene promoters of the R genes may not be as highly expressed in soybean (Arabidopsis and soybean have evolved divergently from one another for over 100 million years, and thus the two species may have diverged enough that promoters are not entirely interchangeable between the two species), and 5) general difficulty of troubleshooting PCR. Nonetheless, we believe that showing expression of three of the four genes in heterologously expressed in soybean demonstrates the ability to leverage the jStack platform for quickly assembling and introducing gene clusters to new species. We have added the following text to the manuscript:

“We then stably transformed the synthetic gene cluster into a new host, soybean (Fig. 3, Supplementary Fig. 3), showing that three of the four R genes were expressed and demonstrating the possibilities of stitching genome-derived gene clusters together for the ultimate purpose of stacking plant resistance traits to various pathogens. Expression of the fourth R genes may not have been detected due to low expression of the genes under our conditions or decreased compatibility of the promoters across species. Future studies leveraging jStack assembly to shuffle gene clusters between species may be most successful between more closely related species, e.g., transferring genomic regions from wild relatives to domesticated species.”

Minor issues:

Lines 148-149 - "We targeted the molecule... This sentence is not clear.

> *Sentenced changed to:*

"As a proof of concept, we set out to produce the molecule bisabolene, a precursor to bisabolane and an alternative to D2 diesel fuel¹⁰."

Lines 153-154 - "First, we tested both cytosolic and plastid expression of bisabolene synthase in tobacco". Where are the results for the individual expression of BisSyn in chloroplasts? This is not shown.

> *We have added measurements of bisabolene from the chloroplast localized BisSyn, and it has equally low production of bisabolene compared to the cytosolic expressed BisSyn. The data is included in Fig 2b.*

Line 186 - "...we assembled a family of nine R gene clusters..." This sentence is misleading, as it seems to indicate that all nine gene clusters were assembled into one plasmid, and this is not shown in the results.

> *We have clarified the sentence by changing it to:*

"To demonstrate how jStack can be used to assemble large genetic regions, we assembled a family of nine R gene clusters scattered across the Arabidopsis genome into nine separate pYB vectors (Supplementary Table 5)"

Lines 241, 243 - NCBI numbers are missing.

> *We have provided all the annotated genbank files alongside with access to request strains and materials through the ICE registry. We also plan to submit the annotated sequences of pYB1301, pYB2301 and pYB3301 to NCBI after the acceptance of the manuscript.*

Line 264 - DH5alpha.

> *changed to "DH5a"*

Line 311 - "...in order for extracted." This sentence does not make sense.

> *Sentence was changed to:*

"One mL of 100% MeOH was added to ground material in for the extraction."

Line 314 - "...each sample..." (not samples); "...together prior to further analysis."

> *The sentence was changed to:*

"Each sample was heated at 80°C and shaken for 15 minutes. Each sample was then spun down"

Line 358 - "...analyzed by microscopy..."

> *Changed to "analyzed by microscopy"*

Line 514 - "...ends are denoted..."

> Changed to “ends are denoted”

Line 517 - "Compatibility of liberated..."

> Corrected to “Compatibility”

Line 533 - "...GFP and DsRed fluorescence."

> Corrected the typo to “fluorescence”

Line 541 - abbreviation for DXP is not defined.

> There is no mention of DXP on line 541

Figure 3a - it would be helpful to explain the criteria for determining the extent of overlap between fragments. Some assemblies have 2 overlapping sequences, while others have 3 or 4. Why is this?

> To clarify the logic in the number of fragments used, we added the following in the main text and legend of Fig. 3 respectively:

“The number of fragments chosen to assemble a whole gene cluster was dictated by the size of the different gene clusters. If a given amplicon did not amplify on the first attempt, the amplicon was into two halves containing an overlapping sequence, increasing the chance of successful PCR due to the smaller amplicon size.”

“Dotted lines represent the PCR amplicons that contain overlapping homologous regions to facilitate yeast homologous recombination into pYB vectors. The number of fragments chosen to assemble a whole gene cluster was dictated by the size of the different gene clusters.”

Reviewer #3 (Remarks to the Author):

The manuscript by Shih et al. describes a new vector system designed for rapid assembly of constructs for heterologous expression in plants. The authors are correct that tools for plant synthetic biology are lacking, and overall, I believe that their system is novel and has the potential to be exceptionally useful. I have some comments and questions that I think should be addressed before publication.

1. The authors describe in detail the overall construction of the vector and the basic strategy of how genes are placed in it. However, while the authors provide an impressive list of 115 parts in Supp. Table 2, very little is said about the compositions/applications of these parts. I believe that this information is provided in their website, though I don't have an account to this website so I haven't checked personally. Could the authors explicitly describe where information about the 115 parts can be found, providing the website in the text.

> As mentioned above, we chose DNA parts to be included in the library based on prior literature describing or characterizing the part. On top of the description of the DNA part, we have included references in Supplementary Table 2 on each part. Moreover, we have written explicit directions on where and how the parts can be found in the Methods section.

2. Some details in the results section regarding the three engineering examples provided would also give a better perspective of the wide range of tools available. In the reporter stacking, bisabolene and violacin production, and resistance gene examples, I did not see what promoter(s) were used in the main text or Methods.

> We have made the figure clearer and added the promoters into the cartoon schematic depicting the genes stacked onto one transfer DNA (T-DNA). We have also added text in the Methods section describing the promoters used to stack the various genes into the final plasmid used. The information for the promoters and the assembly of the final constructs are more explicitly described in Supplementary Table 3. We hope that the updated figure makes clear that the genes were stacked into one plasmid.

I was also not clear from the text if stacking always meant stacking of genes in the vector, or stacking of multiple vectors, each containing a single gene. For example, for violacein, were five agro strains, each harboring one gene, or one construct harboring all five genes? These details are important and would provide a much clearer picture of the system.

> The term 'gene stacking' can mean different things to various researchers. We agree that a clearer and upfront definition of the term 'gene stacking' needs to be included in the manuscript. We apologize for any confusion and have added the following text to clarify the definition our 'gene stacking' within the contexts of this study:

“The term ‘gene stacking’ may refer to a number of strategies to assemble combinations of genes/alleles together, such as 1) crossing/breeding two traits from separate parent strains into one line, 2) successively transforming single genes into one host to introduce multiple genes in a piecemeal approach, and 3) transiently expressing multiple genes using an Agrobacterium-mediated transformation method where multiple Agrobacterium strains are used and each strain contains plasmids to express a single gene. In this study, we refer to gene stacking as assembling multiple gene cassettes into the transfer DNA region (T-DNA) of a binary plasmid to simultaneously deliver multiple genes into a single locus in the host-plant genome in one transformation event, enabling cleaner transgenic events where all transgenes are physically linked together.”

> All five genes necessary for violacein production were assembled into one plasmid in one strain of Agrobacteria. We have changed Figure 2 to show the promoters between all the genes to more clearly represent that all the genes were stacked into one T-DNA. We have also added text to the Methods section describing the specific promoters used to stack all the genes together into one T-DNA.

3. With the violacein pathway, the supplementary data shows that a couple of biosynthetic intermediates accumulate in addition to the final product. While it is beyond the scope of this manuscript to optimize this pathway, some minimal discussion of the results of the engineered pathway- such as what intermediates accumulate, and possibly why they do- and also some mention of how the power of this expression system could be used to optimize the production would be valuable.

> We have added the following text touching upon the results of violacein intermediates:

“The accumulation of the 2-Imino-3-(indol-3-yl)propanoate (IPA imine) intermediate (Supplementary Fig. 2) suggests that the VioB enzyme may be the main rate-limiting step in this

violacein pathway. Future efforts may leverage the library of promoters and jStack assembly platform to test various combinations of gene cassettes to improve upon the yields of violacein in planta.”

4. An exciting part of the manuscript describes the cloning of plant genomic regions, or clusters. The authors showed that two R gene clusters could be stitched together and transformed into soybean hairy roots. The authors state that at least one of the R genes was expressed as shown in Figure 3. Why was the expression of only one gene tested, when, as I understand the figure, 4 R genes were placed into the vector?

> We apologize for the confusion as to the main point of this section. As abovementioned, we wanted to draw attention to the point that we could assemble synthetic gene clusters and shuffle them to new plant species. We focused on only one gene in order to show a proof-of-concept that the gene cluster could be functional in a heterologous species. In response to the Reviewer’s concern, we have attempted RT-PCR on the remaining three genes. We have included data for expression of three of the four genes in the synthetic gene cluster. We encountered difficulty troubleshooting the RT-PCR for the last gene (At3g51570) for a number of possible reasons: 1) this R gene is expressed at very low level, 2) it is not expressed under this growing conditions, 3) it is repressed by the Agrobacterium genes present in the RiT-DNA, 4) the Arabidopsis R gene promoters of the R genes may not be as highly expressed in soybean (Arabidopsis and soybean have evolved divergently from one another for over 100 million years, and thus the two species may have diverged enough that promoters are not entirely interchangeable between the two species), and 5) general difficulty of troubleshooting PCR. Nonetheless, we believe that showing expression of three of the four genes in heterologously expressed in soybean demonstrates the ability to leverage the jStack platform for quickly assembling and introducing gene clusters to new species. We have added the following text to clarify and explain the purpose of this section as well as the nuances of shuffling genomic regions between evolutionary distant species:

“We then stably transformed the synthetic gene cluster into a new host, soybean (Fig. 3, Supplementary Fig. 3), showing that three of the four R genes were expressed and demonstrating the possibilities of stitching genome-derived gene clusters together for the ultimate purpose of stacking plant resistance traits to various pathogens. Expression of the fourth R genes may not have been detected due to low expression of the genes under our conditions or decreased compatibility of the promoters across species. Future studies leveraging jStack assembly to shuffle gene clusters between species may be most successful between more closely related species, e.g., transferring genomic regions from wild relatives to domesticated species.”

5. Some discussion of the limits of the insert size would also be helpful, especially for the non-specialist. Notably, the gene clusters that encode secondary metabolites are much larger than the 10-20 kbp used for the R genes.

> As abovementioned, we have added a section describing the limits of insert size in the text. The following text was added:

“Although there is undoubtedly an upper limit to the amount of DNA that can be transferred from via Agrobacterium mediated transformation, previous studies have transferred bacterial artificial chromosomes (BACs) into plants from binary vectors. Liu et al. reported only a slight decrease in transformation efficiency by 37% when comparing the transfer of an 80 kb versus a 4.4 kb T-DNA via Agrobacterium²⁰. Other studies have reported the transfer of T-DNA as large as 150 kb²¹. Although our study does not assemble anything over 20kb, we envision that most plant engineering efforts will fall well under total size of reported large T-DNAs.”

Reviewer #4 (Remarks to the Author):

In the manuscript of Shih et al., the authors present a method called jSTACK to assemble multiple DNA fragments making use of yeast homologous recombination (HR). They tailored a regular plant binary vector into a so-called trinary vector for extra plasmid propagation in yeast, and designed a two-tier strategy to streamline DNA assembly. They demonstrated a few examples of using jSTACK to stack 2-5 gene expression cassettes into a trinary vector and expressing these genes in *Nicotiana benthamiana* leaves or soybean roots via *Agrobacterium*-mediated transformation. The jSTACK method is undoubtedly a good complement to current DNA assembly tools for plant synthetic biology research.

Major concerns:

1. The advantage of jSTACK over the Gibson assembly (GA) has not been addressed. Although yeast HR can facilitate multiple DNA assembly, the jSTACK method requires yeast and a customized trinary vector as additional experimental materials that are not readily available in plant research laboratories. In contrast, GA can be conducted with homemade or commercially available reagents in a cell-free system.

>We agree that GA can be conducted with homemade or commercially available reagents; however, yeast assembly also can be performed with homemade yeast competent cells as well as plasmid recovery (Loque et al 2007, Nature 446(7132): 195-198). Furthermore, the difficulty in assembly increases with larger fragments and increased number of pieces of DNA. For many synthetic biology studies that have attempted or begun working on larger assemblies, yeast assembly has reliably become a top pick. For example, Daniel Gibson (the inventor of Gibson Assembly) has used yeast assembly to assemble for larger fragments of DNA to stitch together a whole bacterial genome (Gibson et al, PNAS 2008; Gibson et al, 2010 Science 329: 52-56; Karas et al Nature methods 10:410-414; Hutchison et al 2016 Science 351: 10.1126/science.aad6253), presumably due to the higher levels of reliability than purely relying on Gibson Assembly for larger assemblies. The study by Gibson and colleagues, inspired us to adopt the yeast assembly platform for plant transformation vectors. Finally, we believe that the two methods are far from being mutually exclusive, as we have demonstrated that our platform is flexible to accept Gibson Assemblies of smaller fragments that can be used to stitch together larger pieces together. We have edited and added to the following text to recognize the ability to incorporate Gibson assembly into jStack and highlight the advantages of yeast assembly for certain projects:

“Although the versatility of jStack permits scientists to freely choose from an array of molecular biology techniques (e.g., PCR amplicons, Gibson assembly, DNA synthesis, endonuclease based cloning) to generate the fragments that will ultimately be recombined into pYB vectors, the importance of having standardized and universal components cannot be understated. In engineering disciplines, the standardization of parts has enabled collaboration through common components and innovation using existing devices. Building on efforts to establish a standard syntax in plant synthetic biology³, we have developed a hierarchical scheme for assembly of genes and DNA parts into pYB vectors. Importantly, our technique is compatible with other existing methods that use Golden Gate cloning^{4,5}, thus maintaining the ability to collaborate and share characterized DNA parts. Importantly, yeast homologous recombination based DNA assembly has proven to be a robust method for large-scale assembly and compatible with other methods^{1,6,7}. Other DNA assembly methods may face challenges when scaling to larger DNA assemblies, as best demonstrated in whole genome synthesis efforts. Moreover, a combination of several methods (e.g., Gibson assembly or endonuclease based cloning) with yeast homologous

recombination based DNA assembly can enable larger assemblies, as Gibson and colleagues leveraged yeast assembly for the final stitching together of an entire bacterial genome, while relying on restriction endonucleases for the construction of smaller DNA fragments¹.”

On the topic of making materials readily available to the plant research laboratories, we have made not only the vectors but also the library of over 100 plant-specific DNA parts readily available prior to publication through the Inventory of Composable Elements- a registry used by synthetic biologists all over the world to catalog and distribute DNA parts. This platform has been already adopted by leading synthetic biology institutions internationally, and we believe making these experimental materials available to other researchers is a key emphasis of this manuscript. We have added the text below in the Methods section more explicitly describing how to get access to our materials in order to facilitate this process.

“All vectors and resources described are publically available and can be found through the Inventory of Composable Elements (ICE) at <https://acs-registry.jbei.org/>. Once logged in, plasmids and strains are listed under the JBEI Public Registry tab.”

2. The authors stacked at most 5 genes in this work, which is not impressive at all, and can routinely be done with the Golden Gate Assembly (GGA) or GA. The authors need to show some cases where gene stacking can be easily accomplished with jSTACK but can barely be done with GGA or GA.

> We agree that Golden gate assembly method can be use to assemble 5 genes as well. Nonetheless, the purpose of the paper is not to emphasize how many genes can be stacked, but rather to emphasize that genes can be stacked in a different manner than Gibson or Golden Gate Assembly and that our jStack method is compatible with most common DNA assembly methods. It is important for scientists to have multiple options to choose from when deciding on a given method that might be best suited for their research endeavors.

Again, the assembly of the synthetic Mycobacterium genome (Gibson et al, PNAS 2008) is a great example of how leveraging different methodologies can be beneficial for different projects. Gibson et al leveraged Gibson Assembly for pieces up to 10kb, which were then used for yeast homologous recombination based DNA assembly to stitch together large DNA parts. Approaching this project by only using Gibson Assembly on this scale would clearly fail, which is why Gibson and colleagues leveraged their DNA assembly capabilities with a yeast homologous recombination based DNA assembly method.

We believe we have already highlighted the strengths of jStack where Gibson and Golden Gate Assembly would prove difficult with our work on cloning out a library of R gene clusters. As abovementioned, within the 9 gene clusters and 1 synthetic gene cluster assembled, a total of 130,566 base pairs were assembled into our pYB vector. The shuffling of large gene clusters is not amenable to gene synthesis for labs without the resources to synthesize tens to hundreds of kilobases of DNA. Furthermore, DNA assembly with Golden Gate cloning would be cumbersome, as one would have to point mutate out all the cut sites, in order to be compatible with the system. If one were to use the MoClo Golden Gate method described by Weber et al 2011, to assemble these gene clusters, one would have to devoid these sequences of all the BsaI, BpiI, and BsmBI cut sites; within the 10 assembled clusters there were 36 BsaI cut sites, 85 BpiI cut sites, and 30 BsmBI cut sites.

3. The prerequisite for the ultimate usefulness of the jSTACK method is that multiple genes stacked by jSTACK can be stably transformed and successfully expressed. Therefore, data

of stable transgenic expression of multiple stacked genes in WHOLE plants (not only in stably transformed soybean roots) are necessary. Moreover, evidence of successful expression of individual stacked genes (e.g., R genes in Fig.3) needs to be provided.

> The purpose of the jStack method is to provide a novel means of DNA assembly into a Ti vector. That being said, we believe we have shown through multiple demonstrations that we can assemble various fragments of DNA together using yeast homologous recombination based DNA assembly method for the transformation of plants. We do not fully understand what would be gained from generating whole plant transformants, as we have already shown the ability of pYB vectors to transform both soybean (stable root transformants) and tobacco (transiently) in this study. It is important to note that the generation of stable transgenic roots using Agrobacterium rhizogenes method is more complex than a standard Arabidopsis transformation since this transformation method requires a double T-DNA transfer to be successful: one of ~ 15 kb derived from the virulent plasmid and the one from pYB vector in which size varies based on the number of gene inserted.

Nonetheless, to address the Reviewer's concern that "stable" transformants might not be generated with our system, we have included unpublished data from our lab using the jStack system to generate stable Arabidopsis transformants. We believe that the data in the manuscript already demonstrates that our suite of vectors can transform plants, and we hope this additional unpublished data will assuage the Reviewer's concerns that perhaps stable transformants can be generated. Below, we have provided expression data obtained by RT-PCR from a separate project outside the scope of this study using jStack to assemble three transgenes (Gene 1, Gene 2, and Gene 3, and an Actin control) into one pYB vector resulting in the synthesis of a T-DNA larger than 16.5 kb in size. Three independent Arabidopsis transformant lines are shown below (columns 1, 2, and 3).

Besides delaying publication of this manuscript due to the long soybean plant transformation process, we do not think that adding the demonstration that we are capable of generating "stable" transgenic soybean will give substantial information since we already demonstrated that developed pYB vectors can be used for transient expression assays, to generate stably transformed roots in soybean, as well "stable" transgenic Arabidopsis plants (whole plants versus roots only)(see provided unpublished data). By providing unpublished data from whole transformed Arabidopsis, we directly address the Reviewer's concerns.

As for evidence for the expression of individually stacked genes, we have already presented data in the manuscript for the stacked reporter constructs, the optimization of bisabolene production, the production of violacein, and expression of three R genes in the stably transformed soybean roots. Furthermore, we have also provided data above from 3 independent lines of stably transformed whole Arabidopsis plants that have used jStack for a different study.

REVIEWERS' COMMENTS:

Reviewer #1 (Remarks to the Author):

My concerns have been addressed by the revisions.

Reviewer #2 (Remarks to the Author):

Shih et al., "A robust gene stacking method utilizing yeast assembly for plant synthetic biology"

This manuscript describes the development of a method to efficiently clone several genes (with their respective transcriptional promoters and terminators) intended for expression in plants or plant cells, which the authors have named jSTACK. The method relies on homologous recombination in yeast cells to achieve higher order assembly of plant transformation vectors.

After both reviewing the authors' Response to the Reviewers and reading the new version of this manuscript, I believe the authors have adequately addressed the concerns I had raised, most importantly regarding the inclusion of metrics of this methodology and data showing that assembly of intended plasmids was successful. Furthermore, with additional insightful suggestions from the other reviewers, the modifications implemented by the authors have made this manuscript significantly better. I do believe this method will be of great utility to the plant synthetic biology community.

In light of the improvements made to this manuscript, I recommend it be published in Nature Communications.

Reviewer #3 (Remarks to the Author):

All of my comments have been addressed. Accept.

Reviewer #4 (Remarks to the Author):

I do appreciate that the jSTACK method can be a useful complement to current DNA assembly methods. However, the authors failed to experimentally demonstrate the full power of jSTACK over current DNA assembly methods (the Reviewer #2 has similar concerns). Instead, they argued "in principle" advantages of jSTACK or referred to the original work by Gibson et al. (SCIENCE 2008, PNAS 2008) of bacterial genome assembly to evidence the elegance of jSTACK. I think the same key idea as Gibson's should be re-published in Nature Communications EIGHT years later (simply because the same DNA assembly strategy is now oriented for plant research?). This work is more suitable to be published by a specialized plant journal.